# Granulocytes and Cells of Granulocyte Origin—The Relevant Players in Colorectal Cancer

**DOI:** 10.3390/ijms22073801

**Published:** 2021-04-06

**Authors:** Izabela Siemińska, Ewa Poljańska, Jarek Baran

**Affiliations:** 1Department of Clinical Immunology, Jagiellonian University Medical College, 30-663 Krakow, Poland; gorskai@wp.pl; 2Laboratory Medicine, Faculty of Pharmacy, Jagiellonian University Medical College, 30-688 Krakow, Poland; ewa.poljanska@gmail.com

**Keywords:** granulocytes, colorectal cancer, myeloid derived suppressor cells, eosinophils, basophils, neutrophils, tumor associated neutrophils

## Abstract

Colorectal cancer (CRC) is one of the most common malignancy and cause of cancer death worldwide, and it still remains a therapeutic challenge for western medicine. There is strong evidence that, in addition to genetic predispositions, environmental factors have also a substantial impact in CRC development. The risk of CRC is attributed, among others to dietary habits, alcohol consumption, whereas physical activity, food containing dietary fiber, dairy products, and calcium supplements have a protective effect. Despite progress in the available therapies, surgery remains a basic treatment option for CRC. Implementation of additional methods of treatment such as chemo- and/or targeted immunotherapy, improved survival rates, however, the results are still far from satisfactory. One of the reasons may be the lack of deeper understanding of the interactions between the tumor and different types of cells, including tumor infiltrating granulocytes. While the role of neutrophils is quite well explored in many cancers, role of eosinophils and basophils is often underestimated. As part of this review, we focused on the function of different granulocyte subsets in CRC, emphasizing the beneficial role of eosinophils and basophils, as well as dichotomic mode of neutrophils action. In addition, we addressed the current knowledge on cells of granulocyte origin, specifically granulocytic myeloid derived suppressor cells (Gr-MDSCs) and their role in development and progression of CRC.

## 1. CRC

CRC is the third most commonly diagnosed malignancy and the second leading cause of cancer death worldwide [1]. The latest data showed also a shift in the mean age of CRC onset from 72 to 66 years of age [2]. Despite the fact that the overall mortality caused by CRC is continuously declining, the survival and quality of life (QOL) remains poor, especially in an advanced stage of disease [3]. Surgery is still a basic treatment option, in particular for early tumor stage, while for more advanced tumors, effective chemotherapy regimens brought improved survival rates, which was additionally enhanced recently by the introduction of monoclonal antibody-based targeted immunotherapy [4]. Currently, tumor histological status and TNM (tumor, nodes, and metastases) staging during diagnosis remains a basic prognostic marker for CRC [5]. Thus, the development of additional systems that can effectively predict the disease’s progression and efficacy of the therapy in individual patients is highly desirable [6]. Age, genetic predispositions and environmental factors, including diet habits—especially red meat consumption, alcohol abuse, and lack of physical activity, among others, play a major role in CRC development. There are also reports documenting association of CRC with obesity, inflammatory bowel disease (IBD) and immunosuppression in organ transplant patients [7], all emphasizing a crucial role of the immune system in CRC development. Although, CRC has long been considered as not typically immunogenic, multiple studies have shown that T lymphocyte component of the tumor mass is relevant and may have a prognostic value for CRC patients [8]. For example, in patients with curatively resected CRC, the tumor infiltrating lymphocytes in the invasive margin and CD8+ T cell in the central part of the tumor were detected, demonstrating an active anti-tumor immune response [9]. However, not all of the T cells in the tumor environment are considered to be beneficial—e.g., regulatory T cells (Treg) have a potent pro-tumorigenic activity [10,11]. Recent data provide new prognostic/predictive modes based on the numbers of tumor-associated macrophages (TAMs) and Treg, being more reliable than traditional indicators for evaluating CRC patients [12]. In this context, the role of granulocytes and their subsets has not been fully recognized. In this review we present the role and mechanisms of action of granulocytes and cells of granulocyte origin, specifically Gr-MDSCs, as possible prognostic factors or therapy targets in CRC.

## 2. Granulocytes

Granulocytes are leukocytes with specific cytoplasmic granules which content determines the type of the Romanowsky staining, allowing to distinguish the main granulocytes subsets, namely eosinophils, basophils, and neutrophils [13].

Eosinophils are characterized by bilobed nucleus and acidophilic cytoplasm which contains large granules with specific eosinophil proteins, including major basic protein (MBP), eosinophil cationic protein (ECP), eosinophil-derived neurotoxin (EDN), and eosinophil peroxidase (EPO) [14].

Basophils are the least numerous granulocytes in the blood. They have characteristic morphology with large, heavily staining granules, containing heparin, chondroitin sulfate, histamine, and other mediators, including neutrophil chemotactic factor (NCF), slow-reacting kallikrein, and platelet-activating factor (PAF) [13].

Neutrophils are the most abundant leukocytes in peripheral blood of healthy adults. They contain typical segmented nucleus and many tiny granules in the cytoplasm [15]. The azurophilic granules are the lysosomes filled with enzymes such as lysozyme, collagenase, and elastase. They contain also suicidal substances including defensins and cathelicidins, playing an important role in killing of pathogens and cancer cells. The other—specific granules are enriched with lactofferin, collagenase, β2 microglobulin, and gelatinase [16]. Neutrophils are short-lived cells—billions of neutrophils are dying every day by apoptosis and are rapidly replaced under steady state conditions [17]. In the case of inflammation (also cancer related) or infections (particularly bacterial), an emergency granulopoiesis occurs, increasing daily production of neutrophils [18,19].

## 3. Tumor-Infiltrating Eosinophils in Prognosis and Immune Response in CRC

In the case of CRC, eosinophils infiltrate to the center of the tumor mass and the invasive tumor front [20]. In this context, an increased number of tumor-infiltrating eosinophils in CRC patients was shown to correlate with prolonged survival and lack of metastasis at the time of tumor resection [21]. A high number of tumor-infiltrating eosinophils is considered as a favorable prognostic factor independent from the tumor stage, its histological grading and vascular or neural invasion [22]. Furthermore, it was proven that the number of tumor-infiltrating eosinophils depends on the malignant potential of the lesion, significantly decreasing with the progression of CRC [23]. The studies by Harbaum et al. distinguishing peritumoral from intratumoral eosinophils showed that only peritumoral eosinophils have been associated with a tumor progression-free survival, independently from vascular invasion and tumor differentiation [24].

Cho et al. in their study focused on the correlation between tissue eosinophilia and the expression of CCL11 and CCL24, the chemokines that enhance the recruitment of eosinophils. They showed that the expression of these chemokines was lower in glandular cells than in stromal cells of the malignant cancer. The authors correlated the decreased expression of chemokines in glandular cells with a low number of eosinophils in CRC tissue and concluded that this mechanism may be used by cancer cells to inhibit the immune response in the tumor lesion [23].

Although, anti-tumor activity of tissue-infiltrating eosinophils in CRC seems to be independent from CD8+ T cells, the most efficient elimination mechanism of tumor cells by eosinophils is associated with their activation by IFNγ [25], suggesting T cells, mucosal epithelial cells, and NK cells as a relevant source of this cytokine [26]. Human eosinophils possess in vitro tumoricidal activity against various cancer cell lines, including bladder carcinoma, melanoma and CRC but not against, e.g., Hodgkin lymphoma cells, suggesting that their anti-tumor action is target specific [27]. In the cytotoxicity against tumor cells, the release of ECP, EDN, TNF, and granzyme A have been involved [28], however eosinophils can induce apoptosis of CRC cells (Colo-205) also through production of reactive oxygen species (ROS) and EPO release [29]. Adhesion to cancer cells, mediated by beta2-integrin LFA-1 (CD11a/CD18) seems to be a critical step for the execution of eosinophils anti-tumor activity [28]. In this context, IL-18 was shown to increase the expression of adhesion molecules involved in cell-to-cell contact—ICAM-1 and LFA-1 on targets and effectors respectively, suggesting that IL-18 is indirectly involved in eosinophil-mediated cytotoxicity [27]. Similarly, IL-25 was documented to possess an anti-tumor activity against various human tumors including CRC in a mouse xenograft model by increasing eosinophil infiltration [30].

In conclusion, tumor-infiltrating eosinophils in CRC were shown to be cytotoxic towards cancer cells and their number in the tumor environment, in a vast majority of the studies, positively correlated with good prognosis for patients.

## 4. Circulating Eosinophils in CRC

Data considering the number of eosinophils in peripheral blood as a parameter in the prognosis for CRC patients are scarce. The first reliable observation showed that tumor infiltration with eosinophils is not necessarily accompanied by blood eosinophilia, although these two conditions may coexist [21]. There is no convincing data, however, linking blood eosinophilia with the severity of the prognosis for CRC patients. On one hand, the reduced level of circulating eosinophils may be associated with shorter overall survival (OS) in stage I CRC patients [31]. In this context, eosinocytopenia has been proposed as an independent risk factor for DFS in stage II and III CRC [31]. On the other hand, no significant association between pretreatment eosinophil blood count and OS of patients with resectable CRC was documented [32].

## 5. Basophilia and the Prognosis in CRC

In many tumors, including CRC, a decrease in blood histamine level is often observed [33]. Further investigation revealed that blood histamine level in patients with primary tumors is decreased due to a reduced number of circulating basophils [34]. On the other hand, allergy is associated with reduced CRC mortality [35,36]. These observations led to the conclusion that the blood basophil count might be associated with the prognosis in CRC. There are only a few articles on this topic, suggesting that a low blood level of basophils before treatment is associated with worse prognosis and higher tumor aggressiveness [31,32,37]. For example, in patients with resectable CRC, pretreatment basophilia correlated with better prognosis during a 5-year observation period (better OS and DFS) [32]. This dependency has even been considered as an independent, poor prognostic factor for OS and DFS in stage II CRC [31]. The positive effect of basophilia in CRC is most likely related to the secretion of basophil granules content, including histamine and proinflammatory cytokines—e.g., TNFα, IL-6, and IL-1β—augmenting inflammatory reaction, recruitment of cancer-specific CD8+ T cells into the tumor and increasing cancer cell apoptosis [38].

In summary, a low level of blood basophils is associated with poor survival of CRC patients, suggesting this parameter as a potential independent prognostic factor. However, due to a very low number of circulating basophils, the real usefulness of monitoring their blood level for clinical oncology has to be further investigated. Up to now, there is no data on the role of tumor-infiltrating basophils in CRC.

## 6. Circulating Neutrophils as a Prognostic Factor in CRC

In the past, neutrophils’ infiltration to the tumor was largely associated with unfavorable prognosis, although evidence for their beneficial effects was also sporadically observed [39]. Just recently, it was observed that the neutrophils to lymphocytes ratio (NLR) might be used as a prognostic marker in CRC, as it sharply increases between stage IIIb and the terminal stage [40]. Subsequent reports confirmed that high NLR is a significant adverse prognostic factor for DFS even at the earlier (II) stage of CRC and for shorter progression-free survival (PFS) [41]. The latter, however, could be overcome by FOLFOXIRI (fluorouracil, leucovorin, oxaliplatin, and irinotecan) and bevacizumab therapy. This data suggests that chemoimmunotherapy is able to counteract the negative prognostic factors, such as NLR in metastatic CRC [42], while surgery itself had no such beneficial effect [43]. Blood neutrophils respond to stimuli, e.g., IL-8, macrophage inflammatory protein 2 (MIP-2) and rapidly migrate to the site of inflammation, where they produce ROS, secrete proteases and release neutrophil extracellular traps (NETs) [44]. Their ability to NETs formation, stimulated by the elevated IL-8 level at the tumor site, was shown to be correlated with CRC liver metastasis [45]. Blockade of NETs formation resulted in decreased tumor metastasis in mice [46,47].

Neutrophils comprise a heterogenous cell population, differing both in their functions and density. In respect to the latter, during separation of peripheral blood leukocytes by Ficoll density gradient centrifugation, normal-density (NDNs) (in the pellet), and low-density neutrophils (LDNs)—located in the interface together with PBMCs [48], can be obtained. LDNs are detected mainly in course of a disease, while NDNs are typical neutrophils, present under both normal and pathological conditions [49]. Recently, it was described that the population of LDNs contains also Gr-MDSCs (see the respective section below).

As most studies related to blood neutrophils in cancer concern either the whole population of circulating neutrophils (NDNs + LDNs) or on LDNs only, data on NDNs are scarce. Very few studies dealing with NDNs in CRC pointed to lower expression of PD-L1 on their surface when compared to LDNs, which may suggest a difference in regulatory functions between the two populations; however, this was not further confirmed by functional analysis [50].

## 7. Tumor-Infiltrating Neutrophils—A Double Edged Sword in CRC

The studies presented so far have focused on neutrophils in patients’ blood, but similar results were obtained when tumor tissues were examined for the presence of tumor associated neutrophils (TANs) [51]. Moreover, it was reported that the number of neutrophils in the center of the primary tumor positively correlates with the formation of tumor deposits and their size [20]. The intratumor neutrophils may also be responsible for the acquisition of metastatic phenotype by the tumor cells [52]. Analyzing the tumor-induced neutrophils recruitment mechanism, it was revealed that TANs are recruited via the CXCL1/8-CXCR2 axis and the loss of SMAD4 [53]. Loss of SMAD4 promotes expression of CCL15 by CRC and recruitment of CCR1^+^ TANs (CCL15-CCR1 axis) with both arginase-1 (ARG-1) and matrix metalloprotease 9 (MMP-9) activity, forming a premetastatic niche for the disseminated tumor cells [54,55]—e.g., in the lungs [56]. Also, CXCR5 and CXCR7 which are additional ligands for CXCR2 recruit CD11b+ MMP9+ Ly6G+ cells in murine CRC model to promote the formation of early metastatic niche [57].

On the other hand, the tumor cells support neutrophil survival and their adaptation to tumor microenvironment, maintaining their pro-tumorigenic functions [58]. Traditionally, regarded as short-lived, neutrophils have been shown to have their lifespans extended in tumor, most likely due to support from tumor secreted cytokines [59,60].

It has to be mentioned that studies on neutrophils infiltrating tumor tissue in CRC are largely inconsistent and despite data suggesting their pro-tumorigenic role, the results documenting the anti-tumor activity of TANs are also available [61]. For instance, histological examination of CRC in stage II revealed that higher TANs infiltrates were associated with increased OS when compared to the patients with low number of TANs [61]. It was documented that neutrophil infiltration is a favorable prognostic factor for the early stages (I-II) of CRC [61] and patients with high tumor infiltration with CD177+ neutrophils had better OS and DFS [62]. This was confirmed also for neutrophils with myeloperoxidase activity [63]. High infiltration of neutrophils at the invasive tumor margin was correlated with reduced distant metastasis, better response to 5-FU-based chemotherapy and overall, more favorable outcome [64]. Moreover, the expression level of CD16 on neutrophils positively correlates with the number of anti-tumor CD8+ and CD4+ T cell subsets [65].

The observed discrepancies in the prognostic value of TANs in CRC may result from the failure of distinguishing between the N1 and N2 polarization status of TANs, having anti- and pro-tumorigenic activity, respectively. The N1 TANs, activated by type I IFNs, inhibit angiogenesis and effectively eliminate tumor cells via antibody dependent cell-mediated cytotoxicity (ADCC) and phagocytosis. On the other hand, N2 TANs inhibit T-cell proliferation (ARG-1 activity) and induce T-cell apoptosis (NO production) [66,67]. TANs contribute also to the tumor invasion and angiogenesis through the production of MMP9 and vascular endothelial growth factor (VEGF) in the primary and metastatic sites [51]. Therefore, it is not surprising that anti-VEGF antibodies have been used in clinical trials of CRC therapy (NCT04715633, NCT00109226). However, some colorectal cancer patients have shown to be refractory to anti-VEGF therapy [68] and the role of neutrophils is suggested [69]. The N1 neutrophils are described as CD54^high^ CD95^high^ cells which produce high amount of TNF, H_2_O_2_, and low levels of IL-8 [51,70,71], whereas the N2 have opposite features. In mouse CRC models, TANs were demonstrated to suppress T cells via TGFβ release, documenting their N2 character [72]. Also, prostaglandin PGE2 released by murine N2 TANs promotes CRC progression, amplifying inflammation and modulating the tumor microenvironment [73]. This is of relevance as TANs play a key role at each stage of CRC, its initiation, proliferation, angiogenesis, and metastasis [74].

In the context of N1 and N2 paradigm in CRC, Triner et al. recently proposed a new concept. The authors suggested that at early stage of tumor development, TANs mainly inhibit the expansion of tumor-associated microbiome and hamper IL-17 dependent tumor development, whereas in established tumors, TANs acquire pro-tumorigenic functions [75]. This was further confirmed by clinical observations, where stage II CRC patients with high levels of TANs had better OS [76].

In vitro polarization of blood-derived primary human neutrophils under a tumor-mimicking environment resulted in obtaining cells with high surface expression of C-X-C motif chemokine receptor 2 (CXCR2) and secretion of high IL-8 amounts [77], typical for N2 TANs. These findings suggest that blood neutrophils can be polarized toward N1- and N2-like cells in the presence of cytokines and soluble factors—e.g., L-lactate, TGFβ, IL-10, PGE2, and G-CSF—present in the blood of cancer patients [77]. This clearly indicates that the polarization status of TANs can be reversed and potentially used in a future therapy. This was already proved by Kalafati et al. reporting that the pre-treatment of mice with β-glucan induced trained granulopoiesis and reprogramming of neutrophils towards anti-tumor population. This process was associated with transcriptomic and epigenetic rewiring, required type I interferon signaling and was independent of adaptive immunity [78].

## 8. Therapies Targeting Neutrophils in CRC

In this context, there are several clinical trials ongoing using the drugs that can affect neutrophil and TAN functions—e.g., phosphodiesterase type 5 inhibitor (NCT02544880), neutrophil elastase inhibitor (NCT01170845), COX2 inhibitor, NCT00752115), and arginase inhibitor (NCT02903914). However, only the last one refers to CRC. Another clinical study refers to aspirin and its protective effect on the development of CRC (NCT02394769). A growing awareness of the neutrophil role in CRC may be further demonstrated by clinical trial focused on evaluation of the postoperative glutamine-dipeptide and/or omega 3 fatty acids supplementation on their activity (NCT01831310). Furthermore, blocking TGFβ involved in the induction of neutrophils with protumor phenotype, was suggested as a potential therapeutic strategy. Indeed, Fridlender et al. showed that blocking of TGFβ increased recruitment of anti-tumor pro-inflammatory neutrophils in mice [71]. Unfortunately, the clinical trials based on this approach failed due to significant side effects as TGFβ is involved in a number of physiological pathways [79]. Another approach is based on the blocking of TGFβ receptor II (IMC-TR1/LY3022859) and despite the promising results in mouse models of CRC and breast cancer [80], it was shown to be ineffective in the later clinical trials [81]. Now LY2157299 and LY3200882 which are the TGFβ receptor type I inhibitors are studied in the advanced CRC (NCT03470350, NCT04031872).

Due to the importance of IL-8 secreted by CRC cells in the recruitment of neutrophils, a good therapeutic approach seems to be blocking the CXCL8 (IL-8) axis [82,83]. Consequently, BMS-986253—an anti-IL-8 antibody, is being tested in clinical trial in combination with Nivolumab (anti-PD-1) in patients with advanced solid tumors, including CRC (NCT03400332). Similarly, blocking granulocyte colony-stimulating factor (G-CSF) and granulocyte-macrophage colony-stimulating factor (GM-CSF) in cancer patients with neutrophilia is also attractive. Both G-CSF and GM-CSF are used in neutropenia related to cancer treatment (NCT04166604, NCT03102606, NCT00002950); however, as was shown in mouse models, both may induce granulocytes with immunosuppressive functions [84,85]. Additionally, both contribute to acquisition by neutrophils pro-metastatic properties [86,87]. However, to optimize the efficacy of anti-tumor immunotherapy a better understanding of neutrophil-related mechanisms in tumor immunity is essential.

The proposed role of TANs and other granulocyte populations in CRC is emphasized in Figure 1.

## 9. Gr-MDSCs—The ‘Bad Guys’ in Anti-Tumor Immune Response

Gr-MDSCs together with monocytic (Mo-MDSCs) and early stage (e-MDSCs) belong to the group of immature myeloid cells with strong immunosuppressive properties [88]. Their level in peripheral blood rises in many pathological conditions, including cancer [89]. Accumulation of MDSCs in cancer is mainly due to the action of inflammatory cytokines released by cells of the tumor microenvironment [90]. MDSCs are capable of suppressing antigen-specific T cell response, their proliferation and IFNγ production [91]. Moreover, their ability to suppress T-cell response can be arranged also by the induction of Treg cells [10,11]. As a result of interactions with other cells, MDSCs manifest their immunosuppressive character and promote tumor progression [92].

Both in the murine models and CRC patients, the level of circulating and tumor resident MDSCs correlate with cancer stage and metastasis development [93]. However, an increased level of MDSCs was also noted in CRC premalignant stage—colon polyposis [94] and IBD, considered as a risk factor for CRC development [95]. Although, there is no agreement in respect to the predominant role of specific MDSCs subset in CRC development and progression [96], the importance of Gr-MDSCs in that process is unquestionable [94,97,98,99,100]. Gr-MDSCs morphologically resemble neutrophils, however they differ from the latter in functions. In particular, Gr-MDSCs are less phagocytic, have higher activity of Arg-1 and myeloperoxidase, produce more ROS and show lower chemotaxis, e.g., toward the supernatants from human cancer cells [101]. Gr-MDSCs possess strong immunosuppressive properties, which normal PMN do not show [91,102]. So far, several mechanisms regulating the action of Gr-MDSCs in CRC have been described [103]. Gr-MDSCs are the primary cells in CRC that express IL-6 in response to inflammation in colon epithelium and local environment, suggesting Gr-MDSCs to be essential for CRC development [104]. Mundy-Bosse et al. confirmed in humans, what was previously observed in mice, the correlation between the CD15+ MDSCs (Gr-MDSCs) and IL-6 level as well as between CD14+ MDSCs (Mo-MDSCs) and IL-10, suggesting that the balance between systemic IL-6 and IL-10 may be responsible for the composition of the MDSCs subsets [100]. In colitis-associated cancer (CAC), it has been shown that GM-CSF plays the main role in the induction of Gr-MDSCs, acting through STAT3 pathway [105]. The role of STAT3 in CRC progression was further demonstrated with respect to T cell suppression by Gr-MDSCs in the CCL2 dependent manner [103]. Role of exosomes, or more general ‘extracellular vesicles’ has also been raised in the Gr-MDSCs functioning in CRC. In a mouse model it was shown that hypoxia and hypoxia-inducible factor 1α (HIF-1α), may induce Gr-MDSCs to secrete exosomes with high expression of S100A9 which may promote CRC growth [106]. Moreover, under hypoxia conditions Gr-MDSCs increase fatty acids uptake and activate fatty acids oxidation (FAO) [107]. This metabolic reprograming results in an increased ability to inhibit T-cell proliferation [107].

Blocking the immunosuppression of MDSCs will benefit antitumor response and improve the efficacy of the immunotherapies. In this context, targeting Gr-MDSCs is a tempting therapeutic option also for CRC treatment. Already the beneficial effects of metformin, which reduces STAT3 phosphorylation and downregulates activity of Gr-MDSCs [108] or inhibition of FAO [107] have been documented; however, other approaches including Gr-MDSCs depletion [109,110], blocking their recruitment [111] and activity [112] have also been taken under consideration and showed promising results [96]. In a mouse CRC model, Gr-MDSCs rapidly underwent apoptosis mediated by tumor necrosis factor related apoptosis-inducing ligand (TRAIL) and its receptor—death receptor 5 [113]. Based on this data, a clinical trial was launched in patients with advanced CRC using a DR5 agonist (DS-8273a) (NCT02991196) [114].

Recently, the relationship between TANs and Gr-MDSCs has been a matter of debate. The major question is if and how Gr-MDSCs differ from TANs. TANs consist of neutrophils infiltrating tumors while MDSCs are accumulated in the tumor-bearing hosts. They are derived from the bone marrow and can be found in peripheral blood and spleen, in addition to the tumor microenvironment. Some authors report that Gr-MDSCs from spleen, bone marrow and blood constitute precursors of TANs [70]. As TANs are a heterogenous population that may have anti- or pro-tumor functions, Gr-MDSCs are more likely the pro-tumor, N2 subset of TANs [115].

In summary, Gr-MDSCs and N2 TANs serve a seminal role in the growth, angiogenesis, metastasis, and invasion in CRC. Their role alongside with other granulocytes and cells of granulocyte origin in CRC is shown in Figure 2. Therefore, these two populations should be considered as future targets in the treatment of CRC.

## 10. Conclusions

Granulocytes constitute a highly heterogenous population in terms of morphology and functions. In case of cancer, they may have both anti- and pro-tumorigenic role. In CRC, eosinophils and basophils seem to be beneficial. High number of eosinophils in peripheral blood as well as at the tumor site positively correlates with patients’ survival. Basophilia is also related to a reduced CRC mortality. In contrast, neutrophils, especially TANs can act both pro- and anti-tumorigenic in CRC. Despite the lack of the suitable markers differentiating N1 from N2 TANs, their polarization status strongly depends on the tumor stage and cooperation with other cells in the local environment. Although, there is no clear evidence on the relationship between TANs and Gr-MDSCs, it has been demonstrated that Gr-MDSCs and N2 TANs serve a seminal role in the growth, angiogenesis, metastasis, and invasion in CRC. Therefore, these two populations should be considered as a future target in the treatment of CRC.

## Figures and Tables

**Figure 1 ijms-22-03801-f001:**
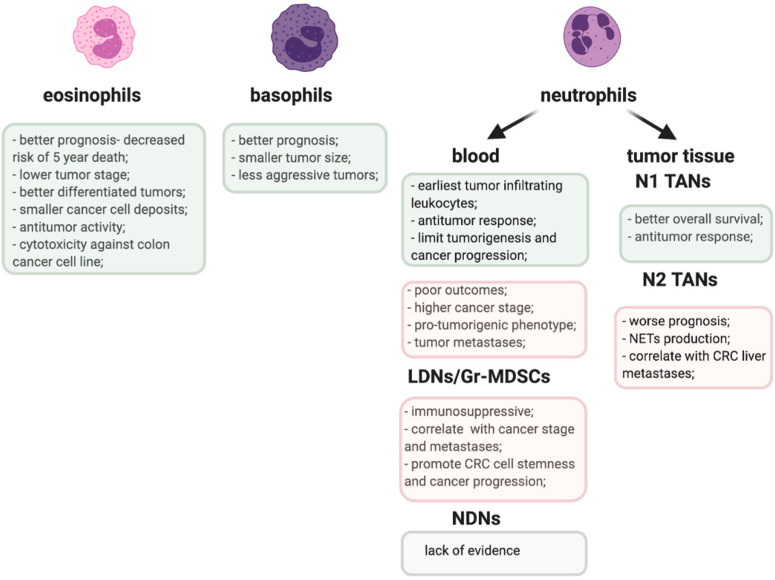
Pro-(pink) and anti-tumorigenic (greenish) activities of granulocyte populations in CRC.

**Figure 2 ijms-22-03801-f002:**
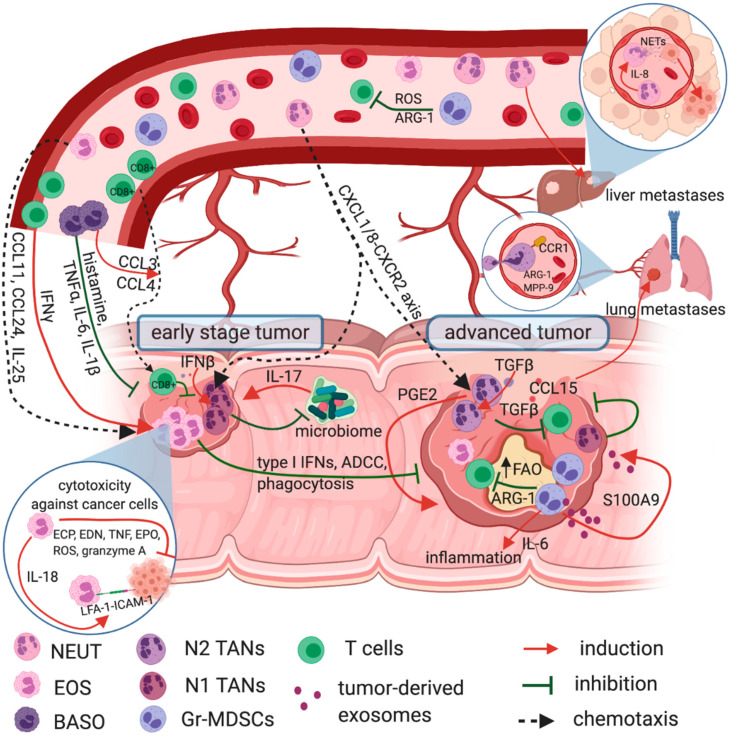
Granulocytes role in CRC. Eosinophils have an anti-tumor role in CRC and their infiltration into the tumor is considered as a positive prognostic marker. They are recruited to the tumor site by CCL11, CCL24, and IL-25 where they are activated by IFNγ to secrete ECP, EDN, EPO, TNF, granzyme A, and produce ROS, all with direct cytotoxicity against cancer cells. Tumoricidal effect of eosinophils depends on cell-to-cell contact and IL-18, which in the autocrine loop enhances the LFA-1—ICAM-1 mediated eosinophil adhesion to the target cells. Basophils are also beneficial in CRC and their role is related to the secretion of the granules content, including histamine and proinflammatory cytokines—e.g., TNFα, Il-6, IL-1β—augmenting inflammatory reaction, recruiting cancer-specific CD8+ T cells (CCL3 and CCL4) into the tumor and enhancing cancer cell apoptosis. Neutrophils possess both anti-tumor and pro-tumorigenic activities. Under the influence of CXCL8 (IL-8) they are recruited from peripheral blood to the tumor site and prompted for NETs formation, which support tumor spread and liver metastasis. Additionally, neutrophils are recruited into the tumor by CCL15-CCR1 pathway, where CCR1+ TANs with ARG-1 and MMP-9 activity are associated with lung metastasis. In tumors, TANs are involved in local immunosuppression through prostaglandin PGE2 and TGFβ secretion, supporting the tumor growth. Unlike N2, N1 TANs possess anti-tumorigenic functions. Their increased infiltration was shown in the early stage of CRC, where they affect the colon microbiome and hamper IL-17 dependent tumor development. Moreover, activated by type I IFNs N1 TANs inhibit angiogenesis and effectively eliminate tumor cells via antibody dependent cell-mediated cytotoxicity (ADCC) or phagocytosis. Gr-MDSCs are potent inhibitors of T cell proliferation and anti-tumor response. Their mechanisms involve ARG-1 activity, ROS production, an increased fatty acid uptake and activated fatty acid oxidation (FAO). Moreover, Gr-MDSCs produce high amounts of IL-6 for sustained inflammation in colon epithelium, thereby promoting CRC progression. Pro-tumorigenic action of Gr-MDSCs also includes secretion of exosomes containing S100A9, which additionally stimulate the tumor growth.

## Data Availability

Not applicable.

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
