# Peer review of "Granulocytes and Cells of Granulocyte Origin—The Relevant Players in Colorectal Cancer"

_ijms, 2021, doi:10.3390/ijms22073801_

Round 1
Reviewer 1 Report
This is a well written review addressing a current topic about the role of granulocytic infiltrations in the pathogenesis of colorectal cancer. However, it would be good if the authors could structure the content taking into account the information described about granulocyte infiltrate in the tumor microenviroment and that found in peripheral blood. It would be easier to read and understand if this information was well structured in the text.
On the other hand, it would be useful and interesting that authors could describe in a little bit more details about the therapeutic approaches that have been developed in based on granulocitic infiltrate in the tumor microenviroments .
Author Response
We thank the Reviewers for the thoughtful and constructive comments. The current version of the manuscript has been revised accordingly and the specific concerns raised by the Reviewers are addressed below and marked with different colors in the main text.
Reviewer 1 (text changes in brown)
- It would be good if the authors could structure the content taking into account the information described about granulocyte infiltrate in the tumor microenviroment and that found in peripheral blood. It would be easier to read and understand if this information was well structured in the text.
We thank the Reviewer for this suggestion. We have structured the text accordingly, taking into account the role of the granulocytes’ specific subsets in CRC in respect to their location either in peripheral blood or in the tumor tissue.
- It would be useful and interesting that authors could describe in a little bit more details about the therapeutic approaches that have been developed in based on granulocitic infiltrate in the tumor microenviroments.
We thank the Reviewer for this comment. The required aspect has been addressed in the section 8 – Therapies targeting neutrophils in CRC.
We hope that we convincingly addressed all the points raised by this Reviewer. Once again, we would like to thank this Reviewer for their comments to improve the quality of our manuscript.
Reviewer 2 Report
The manuscript ¨Granulocytes and cells of granulocyte origin – the relevant players in colorectal cancer¨ addresses comprehensively the key roles of granulocytes and related cells in CRC development. The review discussed deeply the last knowledge on granulocyte involvement in CRC and provides readers with an invaluable guide to further investigate in the field. The manuscript is ready for publishing after two minor issues:
- Did the authors find the relevance of intratumoral basophils?
- It would be advisable to describe at least in one paragraph-before section 2- the granulocyte family for readers not familiarized with the field to get basic knowledge before jumping into each type of granulocyte.
Author Response
We thank the Reviewers for the thoughtful and constructive comments. The current version of the manuscript has been revised accordingly and the specific concerns raised by the Reviewers are addressed below and marked with different colors in the main text.
Reviewer 2 (text changes in green)
- Did the authors find the relevance of intratumoral basophils?
We thank the Reviewer for this relevant question. However, to the best of our knowledge, up to now there is no reports on the role of tumor-infiltrating basophils in CRC (although existing for other types of tumor, e.g. lung cancer or ovarian cancer). The clarification has been introduced in the main text (line 152-153).
- It would be advisable to describe at least in one paragraph-before section 2- the granulocyte family for readers not familiarized with the field to get basic knowledge before jumping into each type of granulocyte.
We have introduced the new paragraph (now section 2 in green) regarding description of the family of granulocytes.
We hope that we convincingly addressed all the points raised by this Reviewer. Once again, we would like to thank this Reviewer for their comments to improve the quality of our manuscript.